# A feasibility open trial of internet-delivered cognitive-behavioural therapy (iCBT) among consumers of a non-governmental mental health organisation with anxiety

Terry Kirkpatrick[1], Linda Manoukian[1], Blake F. Dear[2], Luke Johnston[2] and Nickolai Titov[2]

[1] The Mental Health Association (MHA) of New South Wales, Sydney, New South Wales, Australia
[2] The Centre for Emotional Health (CEH), Department of Psychology, Macquarie University, Sydney, Australia

Corresponding author
Blake F. Dear,
blake.dear@mq.edu.au

## ABSTRACT

**Background.** To date the efficacy and acceptability of internet-delivered cognitive behavioural treatments (iCBT) has been examined in clinical trials and specialist facilities. The present study reports the acceptability, feasibility and preliminary efficacy of an established iCBT treatment course (the *Wellbeing Course*) administered by a not-for-profit non-governmental organisation, the Mental Health Association (MHA) of New South Wales, to consumers with symptoms of anxiety.

**Methods.** Ten individuals who contacted the MHA's telephone support line or visited the MHA's website and reported at least mild symptoms of anxiety (GAD-7 total scores $\geq 5$) were admitted to the study. Participants were provided access to the *Wellbeing Course*, which comprises five online lessons and homework assignments, and brief weekly support from an MHA staff member via telephone and email. The MHA staff member was an experienced mental health professional and received minimal training in administering the intervention.

**Results.** All 10 participants completed the course within the 8 weeks. Post-treatment and two month follow-up questionnaires were completed by all participants. Mean within-group effect sizes (Cohen's *d*) for the Generalized Anxiety Disorder 7 Item (GAD-7) and Patient Health Questionnaire 9 Item (PHQ-9) were large (i.e., > .80) and consistent with previous controlled research. The Course was also rated as highly acceptable with all 10 participants reporting it was worth their time and they would recommend it to a friend.

**Conclusions.** These results provide support for the potential clinical utility of iCBT interventions and the acceptability and feasibility of employing non-governmental mental health organisations to deliver these treatments. However, further research is needed to examine the clinical efficacy and cost-effectiveness of delivering iCBT via such organisations.

## INTRODUCTION

The World Mental Health Surveys indicate that more than 500 million people meet diagnostic criteria for anxiety and depressive disorders each year globally (*Kessler et al., 2009*). Considerable research now confirms that effective psychological treatments exist for these disorders (*Butler et al., 2006*) but that many people experience difficulty accessing traditional face-to-face mental health services (*Burgess et al., 2009*). Reflecting this, effective psychological treatments are increasingly being made available for these conditions via the internet (internet psychotherapy; iPT) (*Andersson & Cuijpers, 2009*; *Andrews et al., 2010*). Internet-delivery of evidence-based treatments offers improved accessibility and convenience for both patients and clinicians (*Titov, 2011*).

To date, the majority of reports have described the results of clinical trials focused on clinical efficacy compared with control groups (*Cuijpers et al., 2009*; *Andersson & Cuijpers, 2009*; *Andrews et al., 2010*). More recently, however, several studies have started to describe the results of computer and internet-delivered cognitive behavioural treatment (iCBT) when used as a part of regular clinical practice (e.g., *Hilvert-Bruce et al., 2012*; *Bergström et al., 2009*; *Learmonth et al., 2008*; *Ruwaard et al., 2012*; *Hedman et al., 2013*; *Carter, Bell & Colhoun, 2013*). For example, one study ($n = 555$ starters) examined the outcomes of Beating the Blues, a computerized cognitive behavior therapy program, when used in an specialist CBT Healthcare Centre in the United Kingdom and found that 25% of program completers achieved reliable and clinically significant changes in both anxiety and depression (*Learmonth et al., 2008*). Another study ($n = 1500$ starters) examined the effectiveness of an online CBT treatment for people referred by GPs for treatment of depression, panic disorder, post-traumatic stress and burnout and found significant improvements in symptoms at post-treatment (*Ruwaard et al., 2012*). Importantly, the results of such trials broadly replicate the results reported in clinical trials, although dropout rates appear higher in some reports or are not clearly described compared with clinical trials (*Hilvert-Bruce et al., 2012*).

Important questions still remain about how these interventions can be disseminated in a way that increases public access to psychological services while maintaining the high standards, clinical outcomes and patient safety seen in clinical trials to date. Converging evidence indicates that internet-delivered psychological interventions can be administered effectively by people with limited mental health training or experience when appropriately trained and supervised (*Titov et al., 2009*; *Robinson et al., 2010*; *Richards & Suckling, 2009*). The non-governmental and community service sector provides a broad free range of services to a large number of people in Australia, including information and advocacy, referral and advice, self-help materials, telephone support as well as often running community-based face-to-face support and treatment programs for consumers, their families and carers. Unfortunately, little is known about the characteristics of people who seek services from this sector and how they might differ from people presenting to traditional health services in Australia (e.g., general practices, mental health treatment in specialist clinics, private psychological treatment). However, a large number of people

are known to utilise the services of the non-governmental and community service mental health sector each year in Australia.

A key strength of the non-governmental and community service sector is that it comprises a large number of paid and volunteer workers and, consequently, represents a large potential workforce who could administer evidence-based internet-delivered interventions to consumers who may not be able or want to access traditional mental health services. Unfortunately, despite the potential, the authors are aware of no studies reporting the use of iCBT by this sector with their consumers.

The primary aim of the present study was to examine the feasibility, acceptability and preliminary efficacy of delivering an established internet-delivered cognitive behavior therapy (iCBT) Course (*Titov et al., 2012*; *Titov et al., 2013*) via non-governmental and community-based mental health organization, the New South Wales (NSW) *Mental Health Association* (MHA), to consumers with symptoms of anxiety. The NSW MHA is a state-based non-governmental mental health organization that provides a very broad range of information, education and support services, via telephone, email and also face-to-face, in New South Wales, Australia, for a large number of consumers, their families and carers. As evidence of its reach, in 2012, the MHA's telephone information and support lines responded to more than 5500 calls and the NSW MHA ran more than 18 separate and continuing self-help support groups across the state (*NSW MHA, 2012*). It was expected that the iCBT Course and approach to delivery would be rated as acceptable and that consumers would report improvements in symptoms of anxiety consistent with clinical trials. Observations by the therapist, who had not previously participated in any previous iCBT interventions, are also reported and it was expected that, based on the little time required, iCBT would be found to be feasible.

## METHOD

### Participants

Participants were visitors to the NSW MHA's website (www.mentalhealth.asn.au) or callers to the NSW MHA's information and telephone support line between July and August 2012. Participants were informed about the study as a trial of online education for symptoms of anxiety via the MHA website and as deemed appropriate in telephone calls.

Interested individuals were sent an Information and Consent Form. Those participants who returned a signed form received a telephone call from an MHA staff member (TK and LM) who conducted a short telephone interview and administered two questionnaires: the Generalised Anxiety Disorder – 7 Item (GAD-7) (*Löwe et al., 2008*) and Patient Health Questionnaire – 9 Item (PHQ-9) (*Kroenke et al., 2010*). The purpose of this interview was to provide further information about the study, answer questions and ensure that participants met the inclusion criteria. Participants were required to meet the following inclusion criteria: (1) Resident of Australia, (2) 18 to 64 years of age, (3) not currently participating in CBT for target symptoms, (4) a score above 4 on the GAD-7 (indicating at least mild anxiety), (5) provides informed consent, (6) access to a computer, printer, and

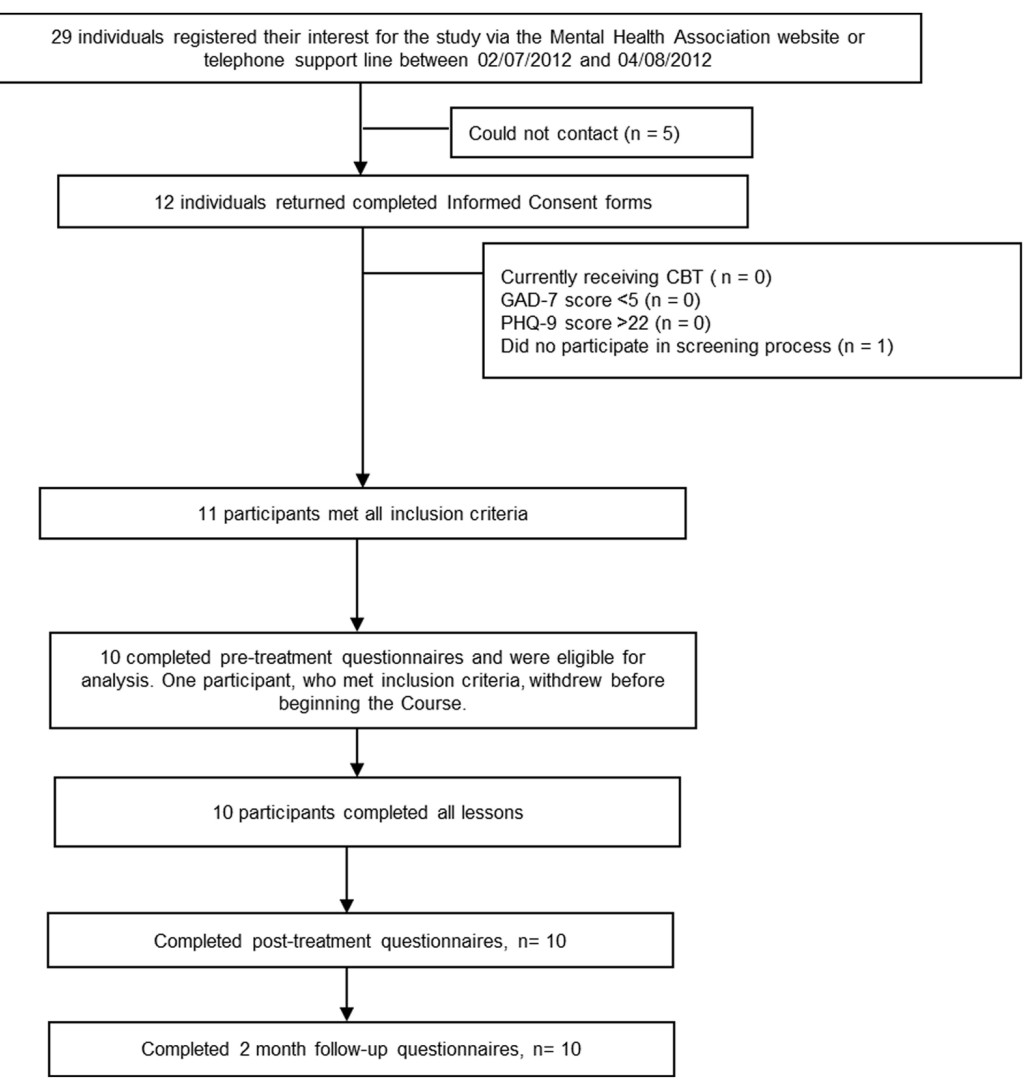

**Figure 1** **Study flow chart.**

internet. Exclusion criteria included (1) severe depression, defined a PHQ-9 total score <23, (2) no suicidal ideation or plan, (3) self-identified as having a principal problem of OCD, (4) acute psychosis.

As indicated in Fig. 1, 29 individuals (20 females and 9 males) indicated an interest in the study, 12 returned consent forms and 11 met all inclusion criteria. One eligible participant requested to be withdrawn from the program before the program began, which left 10 (6 females and 4 males) participants eligible for analysis. Two participants were aged between 25 and 34 years, 5 participants were aged between 35 and 44 years, and 3 participants were aged between 45 and 54. Moreover, 2 participants identified were located in the Sydney metropolitan area and 6 identified as being from regional New South Wales. Two participants were from interstate. No further demographic details were gathered.

## Study design

A single group open trail design was employed to generate data about the preliminary efficacy, safety and acceptability of the MHA providing iCBT for its consumers. Participants were administered all of the questionnaire outcome measures at pre-treatment, post-treatment and 2 month follow-up. A sample size of 15 was calculated as sufficient (one-tailed test, power at 80%, and alpha at 0.05) to detect within-group differences in effect size of 0.7, which was considered the minimum likely effect based on previous studies employing the Wellbeing Course (*Titov et al., 2013*).

This study was approved by the Human Research Ethics Committee of Macquarie University, Sydney, Australia, and registered with the Australian and New Zealand Clinical Trials registry as ACTRN12612000832875.

## Outcomes

Treatment acceptability was assessed at post-treatment via 4 questions: (1) 'Overall, how satisfied were you with the Course?', (2) 'How satisfied were you with the Lessons and Lesson Summaries?', (3) 'Would you feel confident in recommending this treatment to a friend?', and (4) 'Was it worth your time doing the Course?' Participants responded to the first two questions using a 5 point Likert scale, which ranged from 'Very Satisfied' to 'Very Dissatisfied' and the second two questions with a simple 'Yes' or 'No' response. These questions have been used in previous research examining the acceptability of iCBT and other similar low-intensity treatments amongst consumers with a range of different conditions and across a range of different age groups (*Dear et al., 2013*; *Wootton et al., 2013*; *Spence et al., 2011*; *Titov et al., 2013*). Treatment feasibility was broadly accessed based on the amount of clinician time involved, combined with the ease with which the MHA could support consumers through the Course, as well as the preliminary outcomes achieved.

The preliminary outcomes were assessed using the GAD-7 and PHQ-9, which both widely used measures based on DSM-IV diagnostic criteria for generalised anxiety disorder and depression, respectively. The Kessler 10 Item (K-10) (*Kessler et al., 2002*) and the Sheehan Disability Scales (SDS) (*Sheehan, 1983*) were employed as measures of general psychological distress and disability, respectively. The GAD-7 has good convergent validity with other anxiety scales, is sensitive to DSM-IV congruent GAD, social phobia, and panic disorder, with increasing scores indicating greater symptom severity (*Kroenke, Spitzer & Williams, 2001*; *Kroenke et al., 2010*). The GAD-7 is increasingly used in research and in large scale dissemination studies as a generic measure of change in anxiety symptoms and a total score of 10 on the GAD-7 is associated with optimal cutpoint to indicate diagnosis of an anxiety disorder (*Richards & Suckling, 2009*; *Löwe et al., 2008*; *Dear et al., 2011b*). A total score of 10 on the PHQ-9 has been identified as an important threshold for identifying DSM-IV congruent depression with increasing scores indicating greater symptom severity, while psychometric studies indicate the measure is sensitive to change (*Kroenke, Spitzer & Williams, 2001*; *Titov et al., 2011a*).

The treatment acceptability questions were administered at post-treatment. The outcome measures were administered online at pre-treatment, post-treatment and 2-month

follow-up. The GAD-7 and PHQ-9 were also administered weekly to assist the therapist in monitoring participants' clinical safety and progress.

## The intervention

The Wellbeing Course was used as the intervention in the present study and it is comprised of 5 transdiagnostic lessons based on models of cognitive behavioural and interpersonal therapies. Recent work indicates that iPT interventions are efficacious when administered by experienced therapists (*Titov et al., 2011b*; *Dear et al., 2011a*; *Titov et al., 2013*) as well as non-therapists under supervision (*Titov et al., 2010*; *Robinson et al., 2010*; *Johnston et al., 2011*).

The Wellbeing Course is based on a pragmatic model of psychotherapeutic change that assumes that symptoms of anxiety and depression are the result of unhelpful habits of thought and actions, that is, maladaptive cognitions and behaviours (*Titov et al., 2013*). This model also assumes that interventions that are structured, systematic and promote adherence and commitment over several months are more likely to facilitate sustained improvements compared with sporadic or unstructured therapy sessions, which may only result in short-term symptom relief.

The Wellbeing Course is, therefore, a highly structured intervention that participants complete over 8 weeks. Participants are strongly encouraged to learn about and practice the psychological skills taught in the course and to adopt these into their everyday lives. The course systematically teaches core psychological skills that aim to reduce the frequency of unhelpful cognitions and behaviours while increasing the frequency of helpful cognitions and behaviours that promote emotional health. Examples of the former include realistic thinking skills, planning, and problem solving skills, assertive communication, behavioural activation and graded exposure. Examples of the latter include patterns of catastrophic and self-defeating thinking, passive or aggressive communication styles, avoidance and behavioural inhibition. The Wellbeing Course was designed as a low intensity intervention, which could be used as a standalone intervention, an intervention for those on waiting lists for traditional therapy, as an adjunct to traditional therapy, or to facilitate treatment gains post-treatment.

The Course material is comprised of both text based instructions and information as well as case-enhanced learning examples. Case-enhanced learning examples are educational stories that identify a problem that are resolved for the learner and they are thought to assist in learning, adherence and engagement while reducing defensiveness (*Wilson & Sherrell, 1993*; *Chang, 2008*; *Hinyard & Kreuter, 2007*). The structure of the Course, content of lessons, and timeline for the release of the additional resources is shown in Table 1. Each lesson is presented in a slide format combining text and photos. Each lesson was comprised of approximately 60 slides and each slide was limited in word count to approximately 50 words per slide. Automated analyses of readability indicate the text has a mean Flesch-Kincaid Grade level of 6.1, a Simple Measure of Gobbledygook (SMOG) Index of 7, and Automated Readability Index of 5.3, indicating suitability for 8 to 9 year olds. Participants are instructed to read lessons in order over 8 weeks.

**Table 1** Means, standard deviations and effect sizes (Cohen's *d* with 95% confidence intervals) on the primary and secondary measures.

| Measures | Pre-treatment mean | Post-treatment mean | 2-month follow-up mean | Pre to post within group effect size | Pre to follow-up within group effect size |
|---|---|---|---|---|---|
| **Primary** | | | | | |
| PHQ-9 | 12.40 (6.08) | 4.40 (2.99) | 2.90 (2.33) | 1.76 (.73 to 2.79) | 2.17 (1.07 to 3.28) |
| GAD-7 | 10.10 (4.43) | 5.50 (4.60) | 4.20 (3.19) | 1.07 (.14 to 2.01) | 1.61 (.60 to 2.62) |
| **Secondary** | | | | | |
| K-10 | 25.60 (7.21) | 17.60 (5.99) | 15.66 (4.55) | 1.27 (.31 to 2.23) | 1.75 (.72 to 2.78) |
| SDS | 16.20 (6.43) | 9.10 (6.19) | 3.70 (4.16) | 1.19 (.24 to 2.14) | 2.43 (1.28 to 3.59) |

**Notes.**

The standard deviations of the means and the confidence intervals of effect sizes are shown in parentheses.

## The clinician and supervision

One psychologist (TK) employed at the MHA provided all clinical contact with participants with basic administrative support from a psychologist-in-training (LM) prior to and after the trial. TK is an experienced psychologist with more than 10 years' experience and clinical training in treating adults with anxiety and depressive disorders. At the time of the trial, TK was a fully licensed psychologist and, as the most senior psychologist at the MHA, was nominated by the MHA to work on the trial. Both TK and LM had no previous experience providing iCBT, although both were familiar with and had training in CBT.

Clinical contact was recorded and occurred via telephone and email. The clinician was provided with a comprehensive eCentreClinic week-to-week clinician manual that outlines what material is available each week and what common questions and issues raised and encountered by participants each week. TK was also provided with template emails, which could be modified and sent to participants if telephone contact was not possible. These email templates inform participants about what is available each week, 'normalise' difficulties practicing the skills and provide information about past participants experiences at the same point in the Course. The clinician received orientation and training in administering the Wellbeing Course by NT and BFD on 4 occasions for a total of approximately 6 hours. The clinician was supervised by BFD and NT throughout the Course on an as-needed basis via telephone and email. BFD and LJ monitored the Course and participant progress from a technical standpoint and BFD sent the clinician an email at the beginning of each, which detailed clinical and administrative issues for the week ahead.

The clinician was advised to limit weekly contact time to approximately 10 min per participant per week unless more time was clinically indicated. Every contact with each participant was recorded as was the total therapist time spent per participant. The clinician aimed to provide the following four components in each interaction with participants: (1) reinforcement of progress, (2) a summary of the key skills described in the program, (3) 'normalising' of difficulties commonly experienced during treatment, and (4) encouragement to continue.

## Procedure

Participants received an email at the start of the Course providing guidelines and a recommended timetable to get the most out of the Course. Participants also received at least two automatic emails per week during the Course. Some emails were triggered based on participant behaviour: specifically, emails were triggered when (1) participants completed each Lesson during the Course, and (2) if participants had not completed a Lesson within 7 days of it becoming available. Emails were also triggered according to the Course timeline: specifically, emails were triggered (1) at the beginning of each week when new Lessons became available or, if no new Lessons became available, to suggest some tasks for the week, and (2) at set times when participants were known to experience increases in symptoms or to have increased difficulties practicing skills (e.g., during the weeks where graded exposure is introduced). The emails were written and designed to (1) make sure participants always new about new content available on the site, (2) remind participants about unread materials, (3) reinforce progress and skills practice, (4) 'normalise' the challenges of learning new skills, and (5) emphasise that explain that symptom reduction required gentle, but consistent, practice of the skills over time. Each email was brief and was comprised of two to three paragraphs containing 3 or 4 concise sentences. Each email used the participant's first name and was written to convey a warm and supportive tone. All participants consented to receive the emails and no emails contained personal or detailed clinical information.

## Statistical analyses

Participant responses to four questions were used as a measure of treatment satisfaction and to assess the acceptability of the overall treatment approach. The assumptions for parametric tests were examined. Changes in outcome measures were evaluated using paired samples $t$-tests. Within-group effect sizes were calculated using Cohen's $d$ calculated on the pooled standard deviations. As described in recent dissemination studies (*Richards & Suckling, 2009*), an estimate of clinically significant improvement or *recovery* was made by identifying the proportion of participants who demonstrated a significant reduction in their symptoms (defined here, as a reduction of 50% of pre-treatment GAD-7 scores). All analyses were performed in SPSS version 19.0 (SPSS, Inc., Chicago, IL).

## RESULTS

### Attrition

Ten of 10 participants (100%) read the five lessons within the 8 week program and post-treatment and follow-up data was obtained from all participants.

### Treatment satisfaction

Participants indicated a high level of satisfaction with the intervention with 10/10 (100%) participants reporting that they were '*Very Satisfied*' with the Course, would '*recommend the treatment to a friend*' who had similar symptoms and that it was '*worth their time*' completing the intervention. Nine of 10 (90%) and 1/10 (10%) reported they were '*Very Satisfied*' and '*Satisfied*' with the Lessons and Lesson Summaries, respectively.

### Number of contacts and clinician time spent in contact

The mean therapist time per participant was 50.60 min (SD = 12.35). This total time was comprised of an average of 24.6 min (SD = 6.75) and 26 min (SD 6.28) per participant for telephone calls and secure private emails, respectively. An additional average 20 to 30 min per patient was required for administrative purposes including the screening telephone call at recruitment. The therapist made a total of 80 telephone calls ($M = 8$; SD = 1.05; range = 7 to 10) and a total of 115 emails to participants ($M = 11.5$; SD = 1.43; range = 9 to 14).

### Adverse events

No adverse events were reported by participants during the trial or between post-treatment and follow-up. No participants reported an increase in symptoms on the outcome measures at any time point.

### Preliminary clinical outcomes and clinical significance

Means, standard deviations and effect sizes for pre-treatment, post-treatment and follow-up are shown in Table 1. Paired samples $t$-tests revealed statistically significant improvements from pre-treatment to post-treatment and from pre-treatment to follow-up for the GAD-7, PHQ-9, K-10, and SDS ($t_9$ range = 2.67 to 7.29, $p$ range < 0.00 to 0.025). Paired samples $t$-tests also revealed statistically significant improvements from post-treatment to follow up on the GAD-7 and SDS ($t_9$ range = 2.87 to 4.17, $p$ range = 0.002 to 0.018), but not the PHQ-9 or K-10 ($t_9$ range = 1.43 to 2.20, $p$ range = .055 to 0.186). These improvements corresponded to large within-group effect sizes across all measures at post-treatment (Cohen's $d$ range = 1.07 to 1.76), which increased in size at follow-up (Cohen's $d$ range = 1.61 to 2.43).

At both post-treatment and follow-up, 9/10 (90%) participants met the criteria for clinically significant recovery on the GAD-7, that is, a post-treatment and follow-up score at least 50% less than their pre-treatment score.

### Therapist's subjective thoughts, observations and experience

The therapist, TK, who had not previously provided online treatments, reported initially being skeptical that online treatment programs for anxiety could be efficacious. While the therapist understood conceptually how cognitive behavioural and interpersonal therapies were provided via the Course, he reported surprise at the level of engagement of the participants and the significant clinical improvements gained from the 8-week Course and maintained at the 2-month follow-up.

Throughout the Course the therapist closely followed the clinician manual limiting participant contact to reinforcing progress and the key skills described in the program as well as normalising the difficulties experienced during treatment and encouraging continuation. Thus, the therapist made a conscious effort to avoid teaching or discussing other skills not covered in the Course. Instead, participants were encouraged to go back over the relevant sections of the program if they experienced any issues rather than receiving additional therapeutic material or skills from the therapist.

Following the experience of facilitating this trial, the therapist would recommend the internet Wellbeing Course for people experiencing distress from anxiety who meet the inclusion criteria and would use iCBT in the future. The therapist reports that, following the trial, he believes the Course could be valuable as a standalone intervention for people living in regional or remote areas who do not have access to mental health professionals and as an adjunct to traditional therapy. He also believes that the online Wellbeing Course has the potential to increase public access to psychological services, particularly those consumers of not-for-profit, non-government organisations offering support for people experiencing anxiety. The therapist, based on the experience of this trial, expects that it may be possible for the Course to be facilitated by appropriately trained and supervised staff, even if they are not registered clinicians or have limited mental health training.

## DISCUSSION

The primary aim of the present study was to examine the feasibility, acceptability and preliminary efficacy of administering iCBT treatment through a non-governmental and community-based organization to clients with symptoms of anxiety. The clinical efficacy of the Course used had been established in previous research (*Titov et al., 2013*) and is widely used in an established specialist online clinical research unit. However, an outstanding question was how the intervention would perform outside of this research unit and when offered by a non-governmental mental health organization as a part of their service. The therapist administering the program had not been involved in online treatment or clinical research trials. However, the therapist received supervision from two clinical psychologists experienced in face-to-face and internet-based administration of cognitive behavioural programs for anxiety and depression.

It was hypothesized that participants would show levels of satisfaction and improvements in symptoms of anxiety similar to those observed in clinical trials and that participants would rate the Course as acceptable. These hypotheses were supported. All participants (10/10) completed the Course within the 8 weeks and large effect sizes (i.e., Cohen's $d > 1.0$) were found across measures of anxiety, depression, general psychological distress and disability consistent with previous research (*Titov et al., 2010*; *Titov et al., 2013*). These effects were maintained at 2 month follow-up and no adverse events were reported. All participants (10/10) reported the Course was worth their time and they would recommend it others, indicating high levels of participant satisfaction and acceptability, consistent with other trials (*Titov et al., 2013*). Importantly, the effect sizes observed in the present study should be treated with considerable caution. Nonetheless, the results of the present trial are promising and provide preliminary evidence of the reliability of iCBT outcomes outside of specialist research clinics and traditional mental health services where structured, evidence-based, treatments are typically provided (*Hilvert-Bruce et al., 2012*; *Bergström et al., 2009*; *Learmonth et al., 2008*; *Ruwaard et al., 2012*; *Hedman et al., 2013*; *Carter, Bell & Colhoun, 2013*). This is significant given the barriers many experience in accessing timely treatment via these means and highlights the potential of the non-governmental and community services as

another possible avenue for increasing access to evidence-based treatment and addressing unmet need for treatment.

Demonstrating the feasibility of iCBT, it is notable that the results of the present study were obtained with relatively little clinician time ($M = 50.60$ min; SD $= 12.35$), which is consistent with other iCBT studies and points to the cost-effectiveness of this approach, especially provided by non-governmental and community services. The reduced amount of clinical time is achieved by relying on the Course to automatically and systematically teach the core skills and provide the necessary information. This then leaves the therapist with time to focus on reinforcing progress, normalizing and troubleshooting difficulties, and encouraging participants. The therapist in this trial had no previous experience with internet-delivered treatments and was provided with regular supervision. The therapist was also provided email templates and a comprehensive week-to-week clinician manual, which outlined the material that was available each week during the Course as well as common clinical questions and issues raised by participants each week. The therapist found the experience of working online satisfying and was surprised by the clinical results obtained with relatively little clinical input.

The present study has a number of important limitations. Firstly, while the results are consistent with previous research (*Titov et al., 2013*), the present study was a feasibility study that employed an uncontrolled design and a relatively small sample. Consequently, it is unclear how robust or how well the results of the present trial might generalize to other or larger samples and, although consistent with previous research (*Titov et al., 2013*), it is possible that any changes observed in symptoms may not have been a result of the iCBT treatment. Secondly, the present study used only one therapist and a common, but relatively short, follow-up period. Thirdly, it focused on individuals with symptoms of anxiety and, although many participants also had symptoms of depression, no formal diagnostic interviews were conducted or information about other contemporaneous treatments was gathered. Consequently, the diagnostic status and diagnostic composition of participants in the trial is unknown and, thus, some caution is needed in interpreting the clinical findings. Further controlled research is therefore needed with larger sample sizes, diagnostic assessments and longer follow-up periods. Future research would also benefit from further exploration of the different kinds of organisations and environments where iCBT could be employed and an exploration of the outcomes associated with the administration of iCBT by a range of health professionals and non-specialists. Previous research has shown that iCBT interventions can be administered effectively by people with limited or no mental health training when appropriately trained and supervised (*Titov et al., 2009*; *Robinson et al., 2010*; *Richards & Suckling, 2009*) and could represent another means of offering iCBT given the large non-specialist workforce of many non-governmental organisations.

Despite these limitations, however, the results of this preliminary feasibility trial are encouraging. They extend recent reports of internet-delivered psychological interventions in primary or secondary care services and examine their use by consumers of non-governmental community-based organisations. The present results point to the

possibility that, with appropriate training and supervision, these organisations may have the capacity to provide internet-based psychological interventions and improve access to evidence-based treatments. However, much more research is needed.

### Funding

This research was enabled by funding from the Australian National Health and Medical Research Council (NHMRC) Project Grant No. 630560. BFD is supported by a National Health and Medical Research Council (NHMRC) Australian Public Health Fellowship. The funders had no role in study design, data collection and analysis, decision to publish, or preparation of the manuscript.

### Grant Disclosures

The following grant information was disclosed by the authors:
Australian National Health and Medical Research Council (NHMRC): 630560.
National Health and Medical Research Council (NHMRC) Australian Public Health Fellowship.

### Competing Interests

N Titov and B Dear are authors and developers of the Wellbeing Course, but derive no personal or financial benefit from it. N Titov and B Dear are funded by the Australian Government to develop and provide a national internet and telephone-delivered treatment service, the MindSpot Clinic (www.mindspot.org), for people with anxiety and depression.

### Author Contributions

- Terry Kirkpatrick conceived and designed the experiments, performed the experiments, analyzed the data, wrote the paper.
- Linda Manoukian conceived and designed the experiments, performed the experiments, wrote the paper.
- Blake F. Dear, Luke Johnston and Nickolai Titov conceived and designed the experiments, performed the experiments, analyzed the data, contributed reagents/materials/analysis tools, wrote the paper.

### Clinical Trial Ethics

The following information was supplied relating to ethical approvals (i.e., approving body and any reference numbers):
Macquarie University Human Research Ethics Committee. MQHREC #5201200348.

### Clinical Trial Registration

The following information was supplied regarding Clinical Trial registration:
Australian and New Zealand Clinical Trials registry #ACTRN12612000832875.

## Supplemental Information

Supplemental information for this article can be found online at http://dx.doi.org/10.7717/peerj.210.

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
