# Peer review of "A feasibility open trial of internet-delivered cognitive-behavioural therapy (iCBT) among consumers of a non-governmental mental health organisation with anxiety"

_PeerJ, doi:10.7717/peerj.210_

## Round 0.1 · original submission · Major Revisions

Dear authors. I have now received two reviews of your interesting manuscript. Both like your ms but provide you with suggestions on how the ms can be improved.

Reviewer 1 ·

Basic reporting

Please see General comments to the author

Experimental design

Please see General comments to the author

Validity of the findings

Please see General comments to the author

Additional comments

To: PeerJ

Review of the paper ”A feasibility open trial of an internet-delivered cognitive-behavioural therapy (iCBT) among consumers of a Non-Governmental Mental Health Organisation with anxiety”.

This paper reports the findings from a study investigating the acceptability of internet-based cognitive behaviour therapy (iCBT) delivered in a not-for-profit non-governmental context. The results showed that the 10 participants found the treatment acceptable and helpful. Significant reductions of depressive symptoms and anxiety were made.

Please find my comments below.

Introduction
1. The main new aspect of this paper is the context in which the iCBT was delivered. I strongly believe that the Introduction would be improved if the authors would describe this setting in more detail. What is different in this setting compare to the conventional health care setting? Who seeks help here, what is the rationale for seeking help here and not going to one’s GP or an outpatient psychiatric clinic? Is this form of health care subsidised just as other Australian mental health care? What particular aspects could potentially be obstacles when it comes to implementing iCBT in t his new setting? In short, a more detailed presentation of the rationale for the study and why it is important is needed.

2. Was it not an aim to investigate efficacy? Judging from the methods used it seems to be. If so, this should be added to the end of the introduction where the aims of the study are presented. If the assessment of anxiety and depressive symptoms and statistical analyses of the changes are just intended as a means of investigating acceptability and feasibility - then “efficacy” should be removed from the first sentence in Methods, section 2.2. Study design.

Methods
3. Outcomes
As investigation of acceptability and feasibility were the main aims of this study, the Methods would benefit from focusing more on these aspects. Was any particular scale used to measure client satisfaction and acceptability, such as the Client Satisfaction Questionnaire or the Credibility scale? How was adverse events assessed? The current Methods is basically outlined as a traditional efficacy paper, again if focus was on acceptability and feasibility, how were these dependent variables operationalized?

4. Therapist
The final sentence of the Introduction states that the therapist had no prior experience of iCBT. The Methods however contains almost nothing on the background of the therapist (except the background as psychologist). As there was only one therapist who delivered the treatments, and as an important feature of the study is whether it is possible to implement iCBT in a new setting without so large investments time wise, I think it is important to provide the reader with information on the therapist’s background. Was it a psychologist specialized in CBT? Was it a licenced psychologist that could work independently and take on full responsibility for the provided treatment? Why was this particular psychologist chosen? Did the psychologist actively contact the research group to become involved or was the therapist assigned this job by his boss at MHA?

5. Participants
The sample seems to comprise participants with anxiety symptoms defined as scoring above a cutoff at the GAD-7 scale. Was there any other form of assessment of diagnostic or subsyndromal status of the participants? That is, is it fair to view this sample as a sample of persons with a principle diagnosis of GAD or subsyndromal GAD? Or could their main problem have been social anxiety disorder, panic disorder or spider phobia for example?

Results
6. It is to some extent a matter of taste, but I would again prefer if findings regarding feasibility and acceptability were presented first in the Results, i.e. before efficacy data. Rationale for this is that these were the main topics to investigate in the study.

7. Therapist experience
The section regarding the subjective experience by the therapist is an essential part of the paper. However, as this is the Results of the paper it needs to be stressed that these are reported subjective interpretations by the therapist. For example, the sentence “The
online Wellbeing Course has potential to increase public access to psychological
services, particularly those consumers of not-for-profit, non-government organisations
offering support for people experiencing anxiety” should either be reported as ”The therapist reported the he believed that ….”. Or the statement is more of a conclusion to be stated in the Discussion.

8. Description of the suitability of analysing the data using parametric tests
As the sample was very small and the potential impact of one or two outliers is large, it would be good if the authors would add information that they have checked for basic parametric assumptions, such as normality, checking for outliers etc.

Discussion
9. The discussion is generally very well written and balanced. As pointed out in it a few prior studies have examined the effectiveness of iCBT in health care settings. Again, I think that the discussion would be improved from making it clearer what the differences are in the present study compare to the earlier studies, i.e. to stress the uniqueness of this study.

10. One interesting result that is not commented upon in the Discussion is the report by the therapist that he believes that the treatment could be delivered by persons who are relatively unqualified as therapists. What gave him this impression? Do the authors believe that it is a good idea to disseminate iCBT using persons without extensive training as therapists and without knowledge about CBT? If so, why? If not, why?

·

Basic reporting

Overall, this is a well-written article that explores dissemination of ICBT in a new way. This is encouraged by the reviewer and while the findings are very preliminary, they are important and lays the foundation for further research.

Some minor comments:
First page: "Andrew et al. 2010" and "Andrews et al. 2011" probably incorrect.

"Bergström" written correct in the references, but not in the text (written as Bergstrom).

Discussion:
- Wouldn't call a within-group effect of > 1.0 "very large"

Table 1:
- Effect sizes below 1 should be written as e.g. "0.73" and not ".73"
- If GAD-7 is the primary outcome measure in relation to PHQ-9, why not write it before PHQ-9 in the Table?

Experimental design

An open trial is a appropriate choice for this kind of pilot trial.

Validity of the findings

Good choice of outcome measures.

The major concern of the study is the small sample size, which is also confirmed by the power analysis conducted by the authors. The authors correctly labels this a limitation. Please discuss this further, i.e. explan WHY "Further comprehensive research is therefore needed with larger sample sizes and longer follow-up periods."

Please also make sure that the conclusions drawn from this study are valid in relation to the very limited sample size.

Additional comments

Important and innovative new step for ICBT dissemination. The program tested is well-established and the research procedures are carried out well. Great work (once again) by the Australian researchers!

---

## Round 0.2 · accepted · Accept

I read your response letter and think you have made all the corrections needed.